# Regulators of Epithelial Sodium Channels in Aldosterone-Sensitive Distal Nephrons (ASDN): Critical Roles of Nedd4L/Nedd4-2 and Salt-Sensitive Hypertension

**DOI:** 10.3390/ijms21113871

**Published:** 2020-05-29

**Authors:** Tomoaki Ishigami, Tabito Kino, Shintaro Minegishi, Naomi Araki, Masanari Umemura, Hisako Ushio, Sae Saigoh, Michiko Sugiyama

**Affiliations:** Department of Medical Science and Cardio-Renal Medicine, Yokohama City University Graduate School of Medicine, 3-9 Fukuura, Yokohama, Kanazawa-Ku 236-0004, Japan; kino-tabio@umin.ac.jp (T.K.); shintaro.minegishi@gmail.com (S.M.); naomiaraki@gmail.com (N.A.); umemurma@yokohama-cu.ac.jp (M.U.); hisahisa5172@yahoo.co.jp (H.U.); saesaigo1019@googlemail.com (S.S.); vn_nv2525smile@yahoo.co.jp (M.S.)

**Keywords:** salt-sensitive hypertension, ubiquitination, Nedd4L/Nedd4-2, epithelial sodium channel, aldosterone sensitive distal nephron, excitation-transcription coupling

## Abstract

Ubiquitination is a representative, reversible biological process of the post-translational modification of various proteins with multiple catalytic reaction sequences, including ubiquitin itself, in addition to E1 ubiquitin activating enzymes, E2 ubiquitin conjugating enzymes, E3 ubiquitin ligase, deubiquitinating enzymes, and proteasome degradation. The ubiquitin–proteasome system is known to play a pivotal role in various molecular life phenomena, including the cell cycle, protein quality, and cell surface expressions of ion-transporters. As such, the failure of this system can lead to cancer, neurodegenerative diseases, cardiovascular diseases, and hypertension. This review article discusses *Nedd4-2*/*NEDD4L,* an E3-ubiquitin ligase involved in salt-sensitive hypertension, drawing from detailed genetic dissection analysis and the development of genetically engineered mice model. Based on our analyses, targeting therapeutic regulations of ubiquitination in the fields of cardio-vascular medicine might be a promising strategy in future. Although the clinical applications of this strategy are limited, compared to those of kinase systems, many compounds with a high pharmacological activity were identified at the basic research level. Therefore, future development could be expected.

## 1. Introduction

Hypertension is a multifactorial disease determined by both genetic and environmental factors, including dietary habit, and is a representative public health issue due to its high prevalence and risk for cardiovascular diseases. The relationship between oral salt intake and elevation of blood pressure were widely observed in clinical trials such as the INTERSALT Study [1], PURE Study [2], and others [3]. The salt sensitivity of blood pressure is defined as “a physiological trait present in rodents and other mammals, including humans, by which the blood pressure (BP) of some members of the population exhibits changes parallel to changes in salt intake” [4]. Genetic analyses for hereditary and familial hypertension by positional cloning in the 1990′s revealed that ion transporters and their accessory proteins, located in the renal tubules are responsible for the development of genetic hypertension [5]. Based on data from our own [6,7,8,9,10,11,12,13,14] and other [15,16,17,18,19,20,21] studies, we previously reported that important factors in the pathogenesis of salt-sensitive hypertension include the mechanism of sodium reabsorption in the renal tubule and the abnormalities that enhance this reabsorption. Furthermore, we also reported that angiotensinogen gene single-nucleotide polymorphisms (SNPs) might explain the pathogenesis of this disease at the levels of molecular genetics, human evolution, developmental engineering, and kidney physiology [7,8,9,11,12,22,23]. Currently, this report describes the already enforced and ongoing investigations into epithelial sodium channels (ENaCs) in terminal nephrons and its regulatory ubiquitinating enzyme, Nedd4L/Nedd4-2, based on our findings, which might provide significant molecular insight into the onset and development of salt-sensitive hypertension.

## 2. Sodium Reabsorption in Terminal Nephrons

Lifton et al. determined that the genes involved in the development of hereditary human hypertension are restricted to the sodium transporter and its associated proteins in the renal tubule [5]. The most important message that can be derived from this finding might be the fact that the first hit in the development of naturally occurring hereditary hypertension in human is due to the enhanced sodium reabsorption mechanism in the renal tubule. We considered that the mechanism of sodium reabsorption in tubules need to be understood to determine the genetic factors involved in salt-sensitive hypertension, i.e., “hypothesis-driven approaches”, which is the purpose of this review.

The function of the kidney is to produce urine by ultrafiltration, during which the primitive urine passes through the renal tubules, and the quality and quantity of the body fluids and electrolytes are maintained through concentration by water re-absorption, appropriate reabsorption of electrolytes, and appropriate secretion of the bicarbonate ions [24,25]. Most sodium ions in the serum are filtered into primitive urine and at the rate according to the nephron segments; >99% of the sodium that is reabsorbed and ultimately filtered in each segment of the renal tubules is excreted into urine. Considering the mechanism that regulates sodium reabsorption, it can be divided into two segments involving the glomerulus-proximal, tubule-loop of the Henle-distal tubule to the macula densa and the tubules distal to the macula densa (connecting tubules, cortical collecting tubules).

During the first part of the process, >90% of the ultrafiltered sodium is reabsorbed. A physiological regulatory mechanism of the renal microcirculation (tubulo-glomerular-feedback; TGF) between the macula densa and the afferent arterioles, detects the chloride ions in the renal tubular fluid flowing into the distal nephron and imports the anions, thus adjusting the diameter of the afferent arterioles. By adjusting the diameter of the afferent arterioles according to the difference between the amount of filtered and reabsorbed sodium chloride (NaCl), the glomerular filtration pressure and amount were adjusted, and the glomerular filtration rate was maintained according to the oral salt intake. A myogenic reaction specific to the afferent arterioles occurs, in which the smooth muscle around the arterioles reflexively contracts due to the blood flow brought to the afferent arterioles, which regulates the diameter of the afferent arterioles. Similar to TGF, this mechanism adjusts the diameter of the afferent arterioles and maintains a constant glomerular filtration pressure and volume. Moreover, the renal tubules distal to the macula densa are collectively called aldosterone-sensitive-distal nephrons (ASDN), and sodium reabsorption is fine-tuned by the regulation of the expression of sodium ion transporters on the apical side of renal tubular cells, mainly through the action of aldosterone, an endocrine factor. Sodium reabsorption at this site comprises <5% of the total reabsorption, but because of an absent or insufficient feedback/compensation mechanism as in TGF, abnormalities about sodium reabsorption at this site is thought to cause blood pressure abnormalities such as the Liddle syndrome or type I pseudo-hypoaldosteronism (type I PHA). Conversely, abnormalities in sodium reabsorption due to irregularities in the ion transporter from the proximal tubule to the macula densa can be canceled and corrected by powerful TGF and myogenic reactions. Even if such irregularities exist in nature, they might not be recognized due to these physiological compensations.

The onset of salt-sensitive hypertension caused by an enhanced renal tubular renin-angiotensin system (RAS) might be caused by abnormalities in the sodium reabsorption mechanism in the terminal nephron, i.e., the ASDN. Our findings in the C57Bl6/J mice showed that angiotensinogen secretion in the proximal tubular cells was enhanced as a result of the enhanced filtered sodium ion [9,11,12], due to excess oral salt intake. We then proposed a model of the onset of salt-sensitive hypertension by paradoxically enhancing angiotensin II stimulation on the apical side of ASDN, in which the likely ion transporter of the angiotensin II action was the epithelial sodium channels (ENaCs) [26,27,28].

The function of renal tubules, that comprise a highly differentiated and complex regulatory system, is difficult to describe in a few words. However, their function can be summarized to facilitate understanding, as follows [29,30,31,32]. The cellular effects of aldosterone are exerted through intracellular mineralocorticoid receptors (MR) that bind not only to aldosterone, but also to the steroid hormone, cortisol, to produce intracellular effects. As the plasma concentration of aldosterone is much lower than that of cortisol, when competing for binding with MR under the same conditions, most MR probably binds to cortisol. Thus, epithelial cells (such as renal tubule cells) that are sensitive to aldosterone express 11-βHSD2 (type 2 11β-hydroxy-steroid dehydrogenase) and cortisol is converted into cortisone, thereby losing its MR binding activity. As a result, aldosterone-specific cellular action can occur by reducing the competitive inhibitory effect of cortisol on aldosterone with regards to MR. MR is expressed in nephrons beyond the early distal convoluted tubule (DCT), but because 11-βHSD2 is expressed only in nephrons after late DCT, the effects of aldosterone are not evident in early DCT. Mineralocorticoid receptors bind cortisol, and the NaCl co-transporter (NCCT) that is sensitive to thiazide diuretics and that is expressed in early DCT, is controlled by aldosterone to a lesser degree. Regulators of NCCT were discovered in the Gordon syndrome (type II PHA) in which hereditary hypertension was thought to be due to abnormal sodium reabsorption through the NCCT as well as abnormalities in WNK1 and WNK4 [33], but details of the mechanism of hypertension onset remain unknown. Through consistent expression of MR and 11-βHSD2, because “cortisol” was inactivated into “cortisone”, an aldosterone was able to exert effects by binding to the MR in the nephrons below the late DCT. Thus, aldosterone regulates the ENaC expression by binding to MR as ASDN, via the aldosterone–MR complex and through the expression of regulatory factors known as AIP (aldosterone inducible protein), such as SGK1 (serum and glucocorticoid-regulated kinase 1), in nephrons after late DCT.

The outline of sodium transport in the ASDN renal tubular epithelial cells is as shown [34] (Figure 1). Na-K-ATPase are expressed on the basolateral side of all ASDN and functions, by ATP-dependently pumping intracellular sodium out of cells and the uptake of potassium into the cell. As a result of this sodium pump and K-ATPase, a transmembrane Na gradient arises, and the sodium is reabsorbed through the ENaCs on the apical side. After intracellular uptake, potassium is secreted lumenally through the action of the potassium channel ROMK. This channel was also under the control of aldosterone, and aldosterone action in the renal tubule appears to act on potassium secretion as well as sodium reabsorption.

## 3. ENaC and Regulation of ENaC Expression in Renal Tubular Epithelial Cells

As ENaC is inhibited by the diuretic amiloride, it is referred to as an “amiloride-sensitive” epithelial sodium channel and is localized in epithelial cells with polarity. That is, it is expressed only in the apical side of the distal tubule, the alveolar epithelium, and colonic mucosa. It comprises *α*, β, and γ subunits, at a ratio of 1α:1β:1γ. The expression of its function requires all three subunits [29,35]. However, the α subunit alone or the combination of αβ and αγ generates a sodium current, albeit incompletely. ENaC molecules have two transmembrane domains comprising an extracellular domain with multiple glycosylation sites, a short NH2 terminus, and COOH terminus, within cells. All three subunits have two proline-rich sequences (P1 and P2) called PY motifs, at their COOH terminal.

The endocrine factors principally regulate intracellular ENaC expression. In addition to aldosterone action through binding to MR, the action of vasopressin (ADH) secreted from the posterior pituitary gland appears to regulate the expression of ENaC via V2 receptor-adenylyl cyclase-cAMP, as well as insulin via the insulin receptor. As it acts on renal tubule cells through the blood stream, ADH acts via a receptor present on the basolateral side [26,27,28]. Abnormalities in the ENaC gene result in diseases such as cystic fibrosis, type I pseudohypoaldosteronism (type I PHA) [36], and Liddle syndrome [37].

## 4. Liddle Syndrome and ENaC

Liddle syndrome is a type of hereditary hypertension described by Liddle et al., in 1963, that has an autosomal-dominant inheritance pattern similar to that of primary aldosteronism, without elevated renin and aldosterone. Patients with Liddle syndrome exhibit hypokalemic alkalosis. The mechanism of this disease was discovered by Lifton et al. in 1995. [38,39] They found a gene mutation common to the P2 region of the proline-rich PY motif at the COOH terminus of β and γ ENaC. In 1996, Staub et al. [40] identified NEDD4 (neural precursor cell-expressed, developmentally, down-regulated 4) as a ubiquitinating enzyme (E3) protein that binds to this PY motif. Subsequent studies showed that NEDD4 ubiquitinates the Lys residue at the NH2 terminus by binding to the PY motif of ENaC, and the ubiquitinated ENaC was internalized into the cells where it was transported to the proteasome and was then degraded [41,42,43]. The post-translational modification and degradation of this protein is critical for the expression of ENaC on the cell membrane surface, and its physiological half-life is a maximum of one hour. A mutation of the PY motif results in impairments of ENaC ubiquitination via Nedd4 inhibition, sustained ENaC expression on the cell surface membrane, and enhanced sodium reabsorption; thus, a model of salt-sensitive hypertension was proposed. Snyder et al. concurrently revealed a similar mechanism [44].

Sodium reabsorption mediated by ENaC in ASDN does not possess sufficient physiological compensatory/feedback mechanisms such as TGF in the upstream nephron and the myogenic reaction in afferent arterioles. Therefore, the sodium reabsorption mechanism fails due to a dysfunctional ENaC-Nedd4 system, whether it is a gain or loss of function, and can eventually lead to abnormal blood pressure, without effective intra-renal correction.

## 5. Background on *NEDD4* and *NEDD4L/Nedd4-2*

The current paradigm of *NEDD4L* and salt-sensitive hypertension was confirmed, based on the following—the discovery and naming of *NEDD4L,* ENaC as a binding factor with respect to Liddle mutation, the discovery of KIAA0439 and *NEDD4L* (*Nedd4-2*), and our discovery of the human *NEDD4L* C2 domain and the V13 G/A mutation [45] (Figure 2). These findings provide a basis for our investigation, as described in the following.

Kumar et al. (1992) studied changes in gene expression during the development of neural precursor cells, using subtraction cloning [46]. A group of discovered genes was named the “neural precursor cell-expressed, developmentally, down-regulated” (*NEDD*) and the fourth among the molecules numbered 1 to 10, in order of identification, was *NEDD4*, which Staub et al. (1996) found to be associated with hypertension [40]. Using the yeast 2 hybrid method, they found that NEDD4 bound to the PY motif where the Liddle mutation of ENaC was concentrated, and that it was a specific ubiquitin ligase of ENaC. Thereafter, *NEDD4* was highlighted, but its significance shifted. With the discovery of the mechanism of ENaC degradation and the failure of retrieval from the cell membrane during the onset of Liddle syndrome, the gene encoding a protein related to this system is notable as a promising candidate associated with salt-sensitive hypertension. However, *NEDD4* was not a potential candidate gene because, new findings came to light, which led to other insights into *NEDD4*/*NEDD4L*.

Ishikawa et al. (1997) sequenced and published 78 unknown cDNA sequences from a human brain cDNA library [47]. Among these, Harvey and Kumar (2001) identified the KIAA0439 gene as having a Hect domain with ubiquitin ligase activity, while analyzing a Nedd4-like gene in a database [48] and found that it could ubiquitinate ENaC. Chen et al. also located a *NEDD4L* gene on human chromosome 18 with a 67% homology to human *NEDD4* [49]. Around the same time, both Staub and Kamynina et al. discovered *Nedd4-2*, which showed a high homology with *Nedd4* (*Nedd4-1*) in mouse cultured cells [31,50]. A comparison of binding between the WW domain and ENaC revealed that NEDD4-2 showed more ubiquitination activity, and that the endogenous ENaC negative regulator was likely to be *Nedd4-2* [50,51,52]. Thus, despite species differences between mice and humans, NEDD4-2 (NEDD4L) was essentially established as the main ubiquitinating enzyme in contrast to NEDD4, which was conventionally regarded as the first ubiquitinating enzyme of ENaC.

While the model of the ENaC-NEDD4 system continued to shift towards NEDD4L/NEDD4-2, we clarified significant differences that led to further findings. We and other authors, compared human/mouse *NEDD4*/*Nedd4* with *NEDD4L/Nedd4-2* and found that *NEDD4* had a C2 domain unlike *NEDD4L/Nedd4-2*, which was a critical difference between the two molecules. However, phylogenetic analyses of gene homology showed that human/mouse *NEDD4L/Nedd4-2* (KIAA0439) has a high homology with *NEDD4* in Xenopus, which has only one *NEDD4*, and the *NEDD4L* gene was thought to have appeared early during the evolutionary process. Based on this finding, the apparent difference between the two genes could not be explained.

## 6. Human NEDD4L Is a Causative Gene of Salt-Sensitive Hypertension

The ENaC-NEDD4L system plays an important role in post-translational modification through ubiquitination, which regulates the ENaC expression on the cell membrane of the terminal nephron. Genetic analyses of the Liddle syndrome and hereditary hypertension found that ENaC, which is restricted to the terminal nephron and regulates sodium reabsorption, and its regulatory protein, NEDD4L, are promising candidate genes (*NEDD4L*) associated with salt-sensitive hypertension. We focused on the importance of the ENaC-NEDD4L system and selected *NEDD4L* as a target in a genetic analysis of salt-sensitive hypertension.

Based on a draft sequence of human chromosome 18 determined using genomic cDNA expression sequence tag (EST) data such as KIAA0439, Xenopus Nedd4 cDNA extracted from GenBank, Human and Mouse EST, and other databases, we performed Basic Local Alignment Search Tool (BLAST) and cross-match analyses. We searched for SNP by analyzing common polymorphisms determined by re-sequencing the genomic DNA of 48 normotensive persons whose DNA samples were registered in the HyperGEN network. We also analyzed human *NEDD4L* gene transcripts using RT-PCR, 5′RACE, and quantitative PCR targeting human RNA. In addition to known exon-introns, including the ATG codon, two exons (1a and 1) for which the splice site on the genomic DNA sequence exon-end was noted and an exon (exon 2) common to the transcription products from both, were discovered in chromosome 18. Exon 2 had a high homology with the C2 domain of the ancestral Nedd4, such as that in Xenopus and fugu and was thought to be a shared C2 domain that was conserved during molecular evolution [45] (Figure 3).

The start codon ATG of KIAA0439 was found in exon 6 of the conventional gene and in exon 7 of the gene that we discovered. Genomic DNA re-sequencing revealed 38 SNP. Among these, the 13th gene mutation (Variant 13 G/A) was found just before the exon 1 splice site. We spliced cDNA synthesized from human RNA through RT-PCR, immediately after the G base in those with a G allele, and the full-length C2 domain after exon 2 was encoded. In contrast, a frame shift occurred in those possessing an A allele because the splice site moved downstream by 10 bases, ending at the stop codon at exon 2. As a result, the gene product of exon 1 was knocked out (Figure 2). The results of the 5′RACE analyses of mRNA from human kidney and adrenal tissues revealed six transcripts. Among these, isoform I was a transcriptional product of exon 1, isoform II was a transcriptional product of exon 1a, and isoform III was located between exons 2 and 3. Exon 2a encoded the transcription initiation site of KIAA0439. Further analysis using quantitative PCR showed isoform I and II expression in the kidney and lung, respectively, suggesting that it might be under tissue-specific transcriptional control [45].

These results showed that the human *NEDD4L* gene expresses two isoforms each with a C2 domain and another without a C2 domain. Among these, isoform I is a gene product resulting from a G/A mutation, a common variant, and isoform I was hypothesized to be associated with salt-sensitive hypertension. Therefore, we studied variant 13 (G/A) in 367 Japanese persons. An analysis based on genotype, alleles, and three inheritance patterns revealed that the GG and G alleles significantly correlated with essential hypertension. We then assessed the significance of the two types of C2 domains in the human Nedd4L gene and six types of transcriptional diversity. The results of a related study of Japanese patients revealed that a variant 13 (G/A) mutation correlated with essential hypertension [53,54,55]. In summary, the human NEDD4L gene is likely to be a causative gene of salt-sensitive hypertension in humans [56,57,58].

## 7. NEDD4L Is the Causative Protein of Salt-Sensitive Hypertension

We investigated the *Nedd4L* gene function in salt-sensitive hypertensive Dahl rats [59]. We initially analyzed the transcriptomes of rat RNA using the 5 ‘RACE method, to determine whether or not the *Nedd4L* gene in rats has a C2 domain. The results revealed an isoform with a C2 domain in rats and a novel exon on rat genomic DNA. The C2 domain of rat *Nedd4L* was 100% identical to the human C2 domain. Northern blotting using a novel “isoform A” possessing a C2 domain and a probe of the WW domain common to other isoforms, revealed the tissue-specific expression of *Nedd4L* gene transcription products, especially of isoform A, the expression of which was localized in the kidneys, lung, brain, and heart [59].

Changes in salt loading were examined by quantitative PCR and in situ hybridization. The results showed that salt intake increased *Nedd4L* expression in normal DR rats, but did not alter *Nedd4L* expression in salt-sensitive DS rats. The trend of *Nedd4L* expression determined by in situ hybridization was similar in kidney tissues [59]. A study of human kidney tissues obtained under written informed consent, showed that *NEDD4L* expression was restricted to late DCT and lower CNT, CCD, and collecting renal tubules, as in rats, and this distribution was consistent with that of ENaC expression, suggesting that *NEDD4L* is involved in the regulation of ENaC expression [60].

Subsequently, we performed in vitro functional experiments using heterologous gene expression systems using Xenopus oocytes. Initially, we successfully cloned three isoforms of human *NEDD4L* both with and without the C2 domain. A significant reduction in the amiloride-sensitive ENaC current by isoform II and III with ENaC cRNA injected was observed, when either isoform II or III cRNAs were injected into the Xenopus oocyte. The current was significantly restored when isoform I cRNAs were coinjected with other isoforms, indicating the dominant negative effects of the isoform I product against the downregulation of cell surface ENaC by isoforms II and III. Such interactions might abnormally increase sodium reabsorption in ASDN, suggesting that the human *NEDD4L* gene, especially the evolutionarily new isoform I, is a candidate gene for salt-sensitive hypertension in human [53,54,60,61].

## 8. Generation of Nedd4-2 C2 KO Mice and Discovery of Salt-Sensitive Hypertension with Potential Contributions to Cardio-Renal Involvements

Lastly, we decided to develop genetic engineered model of salt-sensitive hypertension such as *Nedd4-2* (*NEDD4L* in human) using knockout mice and examined the detailed phenotypic manifestations to determine the critical roles of *Nedd4-2* in salt-sensitive hypertension. First, we determined genetic variations of mice *Nedd4-2* using *in silico* exploration of the transcriptional start site of the gene. As reported previously, both human and rodent *NEDD4L*/*Nedd4-2* showed molecular diversity, with and without a C2 domain in their N-terminal. The *NEDD4L*/*Nedd4-2* isoforms with a C2 domain were hypothesized to be related closely to ubiquitination of ENaCs. *In silico* gene identification analysis of mice *Nedd4-2* C2 domain coding exon was performed using the BLAST and cross-match program. Then, the sequences of EST and cDNA in the GenBank database (nr) were aligned and compared with the sequence of mice *Nedd4-2* exon 4, which was already known as the transcription start site of mice *Nedd4-2* without a C2 domain. Cross-match analyses were performed repeatedly between the cDNA/EST sequences and genomic sequences, and the results were parsed with chromosome 18 of C57Bl6/J, to form a consistent assembly of EST, cDNA, and a genomic sequence. Thus, the expressed sequence tag and cDNA alignment and the results of the *in silico* bioinformatic database analysis of *Nedd4-2* on mouse chromosome 18q showed a newly identified exon2 coding the C2 domain of mice *Nedd4-2* [62].

Subsequently, we started to create a targeting vector for the newly discovered “exon 2”, which codes the C2 domain of NEDD4-2 and generated genetically engineered mice without a C2 domain of NEDD4-2 [62]. Mice without a NEDD4-2 C2 domain did not show any growth retardation or infertility. We performed a detailed metabolic balance study and continuous blood pressure monitoring analyses, using metabolic cages for individual mice and unrestrained telemetry systems. Under normal oral salt intake, both the wild littermates and *Nedd4-2* C2 KO mice did not show any phenotypic differences, according to blood pressure, urinary sodium excretion, urinary osmotic pressure, urine volume, and water intake. However, under high oral salt intake, *Nedd4-2* C2 KO mice showed blood pressure elevation with reduced urinary sodium excretion [62]. Detailed quantitative analyses for the mRNA expressions levels along laser-captured urinary tubules showed a significant step-wise elevation of ENaC mRNA expressions, in accordance with both their genetic background and oral salt intake, which was abolished into the normal, through amiloride treatment [62]. ENaC mRNA expressions were paradoxically enhanced, despite a high oral salt intake, with a condition of single exon ablation for *Nedd4-2* gene. These results suggested that ENaC itself act as a “sensor” of intra-tubular salt with intra-tubular epithelial “excite-transcription coupling”, which might regulate ENaC gene expressions paradoxically without the NEDD4-2 C2 domain [62]. This could be a pathological molecular mechanism underlying the salt-sensitive hypertension.

As NEDD4-2 with C2 domain is expressed not only in renal tubular cells but also in cardiomyocytes working as ubiquitinating enzyme for voltage-gated sodium-channel, SCN5a, we performed subsequent experiments examining the electrophysiological change of the heart after myocardial infarction, in mice. Additionally, to determine detailed characteristics of salt-sensitive hypertension of the mice, we tried a mineral corticoid receptor antagonist treatment for *Nedd4-2* C2 KO mice. Finally, us and other authors [63,64], have previously found that *Nedd4-2* C2 KO mice showed eplerenone-resistant salt-sensitive hypertension [65] and enhanced electrophysiological abnormalities, after myocardial infarction [66], suggesting a pivotal role of the *Nedd4-2* isoform with C2 domain for cardio-renal association, with regards to target-organ damages of the subjects with salt-sensitive hypertension.

## 9. Summary

In summary, from earlier detailed re-sequencing experiments for the human *NEDD4L* gene [45], we performed gene targeting experiments for the newly discovered exon, which encoded a C2 domain of mice Nedd4-2 [62]. Other experiments were performed including the discovery of a rodent *NEDD4L* gene C2 domain expressed along urinary tubules [59], heterologous gene expression in the Xenopus oocytes experiments with dominant negative effects of newly discovered human isoform I [60], the discovery of the NPC2 protein as a C2 domain binding protein in urinary tubules [61], and a genetic association study for human hypertension [53,54,55]. Ultimately, we found that the lack of single isoform of the gene caused a significant change *in vivo*, resulting in a higher oral salt intake suppressing sodium excretion in urine and elevated blood pressures, the pathophysiology of which is called “salt-sensitive” [62]. In accordance with human genetic studies, the impairment of tubular sodium transport might be pivotal in the onset and development of salt-sensitive hypertension [65,66].

The author’s studies of *NEDD4L* were derived from a collaboration between 2000 and 2003 with Professor Robert Weiss at the Eccles Institute of Human Genetics at the University of Utah. Professor Robert Weiss is a bioinformatics expert who also participated in the Human Genome project, and some of the results described herein were derived from *in silico* analyses. This is an example of the ability to discover valuable information from vast databases using a desktop computer.

The Human Genome project was completed in 2003 which provided many insights, such as the fact that only 30,000 to 40,000 genes are needed to create diverse humans, instead of the original prediction of 100,000 to 120,000 genes. We found that one *NEDD4L* gene produces multiple transcripts. Thus, many genes might generate and maintain biological diversity via transcriptional diversity. The *NEDD4L* gene is also notable as an example of transcriptome analysis after genomic and proteomic analyses.

Human *NEDD4L* might be the next candidate gene for salt-sensitive hypertension, following angiotensinogen. Although there is currently insufficient evidence to fully support this hypothesis, when the amount of information available about the *NEDD4L* gene becomes similar to that of the angiotensinogen gene, the significance of the *NEDD4L* gene in the onset and progression of salt-sensitive hypertension will surely be further clarified by investigating its functions in vitro, in vivo, and clinically. Furthermore, therapeutic and diagnostic applications targeting the *NEDD4L* gene should become available in the foreseeable future, including de-ubiquitination enzyme activating agents for antagonizing human *NEDD4L.*

## Figures and Tables

**Figure 1 ijms-21-03871-f001:**
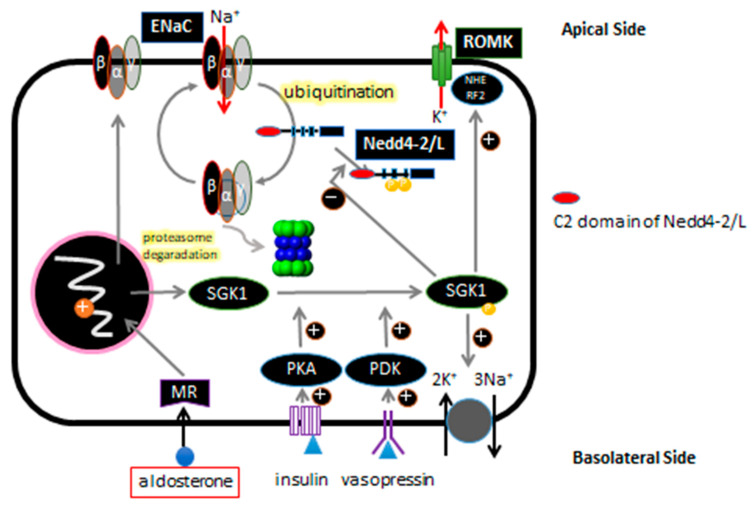
Schematic presentation of the ASDN epithelial cell is shown. ENaC is ubiquitinated by Nedd4-2/Nedd4L, and subsequently degraded by proteasome. Aldosterone, insulin, and vasopressin regulate ENaC gene expression via the basolateral side.

**Figure 2 ijms-21-03871-f002:**
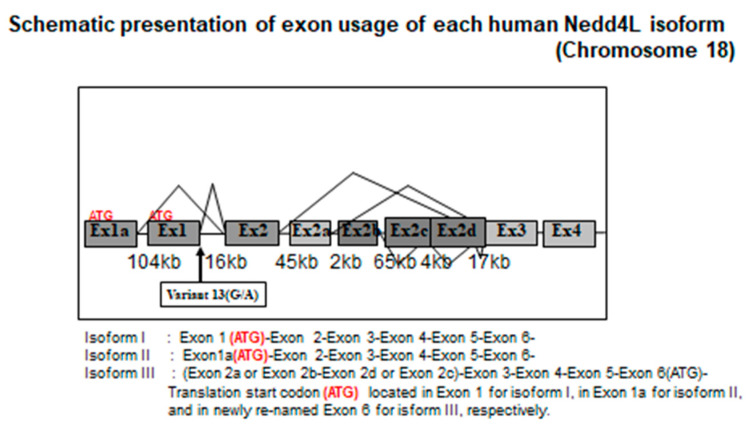
Schematic presentation of exon usage of each human Nedd4L isoform is shown. Exon in dark square shows the newly discovered exons through re-sequencing. Variant 13 (G→A) located at the end of the newly discovered exon 1. Translation start codon (ATG) located in Exon 1 for isoform I, in Exon 1a for isoform II, and newly re-named Exon 6 for isoform III, respectively. (modified from [45]).

**Figure 3 ijms-21-03871-f003:**
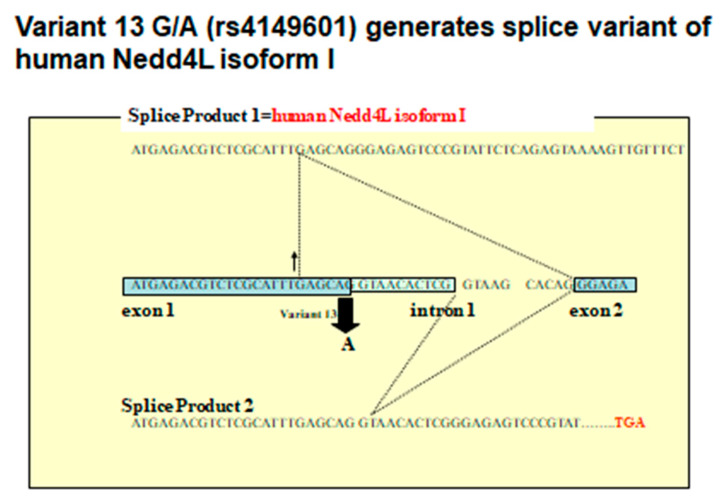
Detailed analyses of variant 13G/A (rs4149601) generating the splice variant of human. Nedd4L isoform I is shown. Splice site moves 10 base pairs and splice product 2 ends in the stop codon in exon 2.(modified from [45]).

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
