# Peer review of "Regulators of Epithelial Sodium Channels in Aldosterone-Sensitive Distal Nephrons (ASDN): Critical Roles of Nedd4L/Nedd4-2 and Salt-Sensitive Hypertension"

_ijms, 2020, doi:10.3390/ijms21113871_

Round 1

Reviewer 1 Report

The authors might include additional text to summarize the major sections of the review.

The authors must carefully check each reference throughout the entire manuscript.  Also, sections of the manuscript are lacking references.

The text of the review should relate directly to the Figure 1 and a notation added to direct the reader to the Figure 1.

The introduction paragraph is not related to the title of the review articles.  The authors should completely revise the opening paragraph to reflect the purpose of the review article.

Abstract.  Line 16.  There is no need to use the word, respectively.

Introduction.  Lines 37, 40.  These reference numbers are incorrect.

Line 54.  The word study should be changed to review.

Line 60-61.  This sentence is incorrect and confusing.

Line 63.  The distal convoluted tubule is distal to the macula densa cells.

Lines 71 and 75.  The word reflux is incorrectly used in these sentences.  The correct term is filtration.

Line 105.  The word below should be replaced with the word beyond.

Lines 113-114.  This sentence is confusing and requires clarification.

Line 120. The word intaking is incorrectly used in this sentence.  The correct wording might be uptake of potassium into the cell.

Line 123.  The word extracellularly should be changed to lumenally.

Line 139.  The word hematogenously is incorrected used in this sentence.

Line 149.  The authors might state that Lifton made this discovery in 1995.

Line 151.  The correct references are 33, 35, and not 34, 35.

Lines 175-176.  This sentence is confusing.

Line 191.  A reference should be added.

Throughout the manuscript the same abbreviation for NEDD should be used (Nedd).

Lines 242, 247.  A reference should be added to both studies.

Lines 294-311.  A reference should be added.

Author Response

Reviewer #1

Comment 1: The authors might include additional text to summarize the major sections of the review.

Response 1: Thank you for your comment. We completely agree with the reviewer, and have added an additional paragraph to the “Summary” section, as follows:

In summary, from earlier detailed re-sequencing experiments for human NEDD4L gene, we performed gene targeting experiments for newly discovered exon which encodes a C2 domain of mice Nedd4-2. Other experiments were performed including the discovery of a rodent NEDD4L gene C2 domain expressed along urinary tubules, heterologous gene expression in Xenopus oocytes experiments with dominant negative effects of newly discovered human isoform I, the discovery of NPC2 protein as a C2 domain binding protein in urinary tubules, and a genetic association study for human hypertension. Ultimately, we found that the lack of single isoform of the gene caused a significant change in vivo resulting in a higher oral salt intake suppressing sodium excretion in urine and elevated blood pressures, the pathophysiology of which is called “salt-sensitive”. In accordance with human genetic studies, the impairment of tubular sodium transport may be pivotal in the onset and development of salt-sensitive hypertension.”

 Comment 2: The authors must carefully check each reference throughout the entire manuscript. Also, sections of the manuscript are lacking references.

Response 2: We would like to take this opportunity to thank you for your time and for providing us with your valuable comments. We completely agree with your opinion. We will revise our bibliography accordingly.

 Comment 3: The text of the review should relate directly to the Figure 1 and a notation added to direct the reader to the Figure 1.

Response 3: Thank you for your comments and for bringing this issue to our attention. According to the reviewer’s comment, we have revised Figure 1 and have added two more Figures, for the reader’s understanding.

Comment 4: The introduction paragraph is not related to the title of the review articles.  The authors should completely revise the opening paragraph to reflect the purpose of the review article.

Response 4: Thank you for your comments. We agree about the excessive description of the tubular renin-angiotensin system in the “Introduction” section. Therefore, we revised this section as follows:

Before: “1. Introduction

Based on data from our own[1-9] and other[10-16] studies, we previously reported that important factors in the pathogenesis of salt-sensitive hypertension include the mechanism of sodium reabsorption in the renal tubule and abnormalities that enhance such reabsorption; we also reported that angiotensinogen gene single-nucleotide polymorphisms (SNPs) can explain the pathogenesis of this disease at the levels of molecular genetics, human evolution, developmental engineering, and kidney physiology[2-4, 6, 7, 17]. In addition to the established (1) systemic renin-angiotensin system (RAS) and (2) local RAS, (3) renal tubular RAS as a third RAS might be significant in the onset of salt-sensitive hypertension. Thus, understanding renal tubular RAS and its functions is indispensable for treating salt-sensitive hypertension in humans[18]. Ultimately, diagnosing individual states of tubular RAS in each patient and identifying therapeutic procedures that could inhibit renal tubular RAS are one of our future goals of translational research in salt-sensitive hypertension. Currently, this report describes already enforced and ongoing investigations into epithelial sodium channels (ENaCs) in terminal nephrons and its regulatory ubiqutinating enzyme, Nedd4L/Nedd4-2, based on our findings.”

After: “1. Introduction

Hypertension is a multifactorial disease determined by both genetic and environmental factors including dietary habit and is a representative public health issue because of its high prevalence and the risk for cardiovascular diseases. The relationship between oral salt intake and elevation of blood pressure have been widely observed in clinical trials such as INTERSALT Study[1], PURE Study[2], and others[3]. The salt sensitivity of blood pressure is defined as “a physiological trait present in rodents and other mammals, including humans, by which the blood pressure (BP) of some members of the population exhibits changes parallel to changes in salt intake”[4]. Genetic analyses for hereditary and familial hypertension by positional cloning in the 1990’s revealed ion transporters and their accessory proteins, located in the renal tubules are responsible for development of genetic hypertension[5]. Based on data from our own[6-14] and other[15-21] studies, we previously reported that important factors in the pathogenesis of salt-sensitive hypertension include the mechanism of sodium reabsorption in the renal tubule and the abnormalities that enhance this reabsorption. Furthermore, we also reported that angiotensinogen gene single-nucleotide polymorphisms (SNPs) may explain the pathogenesis of this disease at the levels of molecular genetics, human evolution, developmental engineering, and kidney physiology[7-9, 11, 12, 22]. Currently, this report describes already enforced and ongoing investigations into epithelial sodium channels (ENaCs) in terminal nephrons and its regulatory ubiquitinating enzyme, Nedd4L/Nedd4-2, based on our findings, which might be significant molecular insight into onset and development of salt-sensitive hypertension.”

 General Comments:

Abstract.  Line 16.  There is no need to use the word, respectively.

Response: Thank you for your comment. We have deleted “respectively” as pointed out.

 Introduction.  Lines 37, 40.  These reference numbers are incorrect.

Response: Thank you for your comment. We have revised the “Introduction” section with regards to the excessive description of tubular renin-angiotensin system.

Line 54.  The word study should be changed to review.

Response: Thank you for your comment. We have changed “study” to “review” in the revised manuscript.

 Line 60-61.  This sentence is incorrect and confusing.

Response: Thank you for your comment. We have revised the language in this section accordingly.

 Line 63.  The distal convoluted tubule is distal to the macula densa cells.

Response: Thank you for your comment. We agreed with your comment and have made the suggested revision:

After: “Considering the mechanism that regulates sodium reabsorption, it can be divided in into two segments involving the glomerulus-proximal tubule-loop of the Henle-distal tubule to macula densa and tubules distal to the macula densa (connecting tubules, cortical collecting tubules).”

Lines 71 and 75.  The word reflux is incorrectly used in these sentences.  The correct term is filtration.

Response: Thank you for your comment. We revised our manuscript accordingly, as follows:

After: “By adjusting the diameter of the afferent arterioles according to the difference between the amount of filtered and reabsorbed sodium chloride (NaCl), glomerular filtration pressure and amount are adjusted, and the glomerular filtration rate is maintained according to oral salt intake.”

“Similar to TGF, adjusts the diameter of the afferent arterioles and keeps the glomerular filtration pressure/volume constant.”

 Line 105.  The word below should be replaced with the word beyond.

Response: Thank you for your comment. We have revised the manuscript according to the reviewer’s comments as follows:

“MR is expressed in nephrons beyond the early distal convoluted tubule (DCT), but because 11-βHSD2 is expressed only in nephrons below the late DCT, aldosterone effects are not evident in the early DCT.”

 Lines 113-114.  This sentence is confusing and requires clarification.

Response: Thank you for your comment We revised the sentences as pointed out by the reviewer, as follows:

Before: “Consistent expression of ENaC and 11-βHSD2, and that cortisol and MR do not exert effects in nephrons below the late DCT, but an aldosterone exerts effects by binding to MR.”

After: “As a result of consistent MR and 11-βHSD2 expression, resulting from the inactivation of “cortisol” into “cortisone”, aldosterone was able to exert effects by binding to MR in nephrons below the late DCT.”

 Line 120. The word intaking is incorrectly used in this sentence. The correct wording might be uptake of potassium into the cell.

Response: Thank you for your comment. We have revised our manuscript as pointed out by the reviewer.

Before: “Na-K-ATPase are expressed on the basolateral side of all ASDN and function by ATP-dependently pumping intracellular sodium out of cells and intaking potassium into the cell. As a result of this sodium pump and K-ATPase, an intracellular sodium gradient arises, and sodium is reabsorbed through ENaCs on the apical side.”

After: “Na-K-ATPase are expressed on the basolateral side of all ASDN and function by ATP-dependently pumping intracellular sodium out of cells and the uptake of potassium into the cell. As a result of this sodium pump and K-ATPase, an intracellular sodium gradient arises, and sodium is reabsorbed through ENaCs on the apical side.”

 Line 123.  The word extracellularly should be changed to lumenally.

Response: Thank you for your comment. We revised our manuscript as pointed out by the reviewer.

Before: “After intracellular uptake, potassium is secreted extracellularly through the action of the potassium channel ROMK.”

After: “After intracellular uptake, potassium is secreted lumenally through the action of the potassium channel ROMK.”

 Line 139.  The word hematogenously is incorrected used in this sentence.

Response: Thank you for your comment. We revised our manuscript as pointed out by the reviewer.

Before: “Because it acts hematogenously on renal tubule cells, it acts via a receptor present on the basolateral side.”

After: “Because it acts on renal tubule cells through the blood stream, ADH acts via a receptor present on the basolateral side.

 Line 149.  The authors might state that Lifton made this discovery in 1995.

Response: Thank you for your comment. We revised our manuscript as pointed out by the reviewer.

Before: “The mechanism of this disease remained obscure until Lifton et al. discovered a gene mutation common to the P2 region of the proline-rich PY motif at the COOH terminus of β and γ ENaC.”

After: “The mechanism of this disease was discovered by Lifton et al. in 1995 when they found a gene mutation common to the P2 region of the proline-rich PY motif at the COOH terminus of β and γ ENaC.”

 Line 151.  The correct references are 33, 35, and not 34, 35.

Response: Thank you for your comment. We revised our manuscript as pointed out by the reviewer.

 Lines 175-176.  This sentence is confusing.

Response: Thank you for your comment. We revised our manuscript as pointed out by the reviewer.

Before: “A group of discovered genes was named “neural precursor cell-expressed, developmentally, down-regulated” (NEDD) and the fourth among molecules numbered 1 to 10 in order of identification was NEDD, which Staub et al. (1996) found was associated with hypertension[36] .”

After: “A group of discovered genes was named “neural precursor cell-expressed, developmentally, down-regulated” (NEDD) and the fourth among molecules numbered 1 to 10 in order of identification was NEDD4, which Staub et al. (1996) found was associated with hypertension[36] .”

 Line 191.  A reference should be added.

Response: Thank you for your comment. We have added this reference according to the reviewer’s suggestion.

 Throughout the manuscript the same abbreviation for NEDD should be used (Nedd).

Response: Thank you for your comment. We corrected the abbreviations for the gene and the protein according to the reviewer’s suggestion.

 Lines 242, 247.  A reference should be added to both studies.

Response: Thank you for your comment. We revised our manuscript as pointed out by the reviewer.

 Lines 294-311. A reference should be added.

Response: Thank you for your comment. We revised our manuscript as pointed out by the reviewer.

Reviewer 2 Report

Reviewing the manuscript entitled, “Regulators of Epithelial Sodium Channels in Aldosterone-Sensitive Distal Nephrons (ASDN): Critical roles of Nedd4L/Nedd4-2 and Salt Sensitive Hypertension.” by Ishigami T. et al., this is a well-written article about relevance between Nedd4, which is an ubiquitinating enzyme, between salt sensitive hypertension. Although this is an important article for further hypertension treatment in the future, the manuscript lacks basic information of what salt sensitive hypertension is. In addition, the lack of information on renal tubular RAS makes it difficult to read at the entrance of this article.

Concerns

First of all, the authors should set up a paragraph on salt-sensitive hypertension and explain it to the readers of this journal.

In 1. Introduction, the authors emphasis the relationship between sodium reabsorption abnormality and renin angiotensin system particular renal tubular RAS for onset of salt-sensitive hypertension. However, the authors do not explain the detailed mechanism of salt-sensitive hypertension. By the way, what is renal tubular RAS? Has it reached to the consensus of all scientists? Authors should explain the phenomenon of generally lacking consensus with a figure.

From line 87 to line 89, the authors mentioned, “The onset of salt-sensitive hypertension caused by an abnormal renal tubular renin-angiotensin system (RAS) might be caused by abnormalities in the sodium reabsorption mechanism in the terminal nephron, that is, ASDN.” What is an abnormal renal tubular renin-angiotensin system (RAS)? First the authors need to mention about the function of normal renal tubular RAS. In addition, in the case of common cardiovascular diseases, RAS is overactive in compensation, and in chronic states, overactivity persists and exacerbates the disease development. This phenomenon should be distinguished from anomalies.

The authors need to describe that detailed mechanisms of regulation of ENaC by the angiotensin II receptor. The authors only described it in line 141 to line 142. This is too short. In your tone of this reviewed article, RAS inhibitors appear to work on salt-sensitive hypertension. However, salt-sensitive hypertension is generally resistant to RAS inhibitors. Based on the clinical findings, you should explain that the therapeutic target for salt-sensitive hypertension is the regulation of ENaC, not the RAS inhibitor using the figure.

Although there is no particular problem was pointed out in the paragraph of Nedd4L / Nedd4-2, if this gene is a candidate for the next salt-sensitive hypertensive drug, it is preferable to describe the specific method of that strategy.

Minor concerns

From line 118 to line 119, the authors mentioned “The outline of sodium transport in ASDN renal tubular epithelial cells is as shown in the reference[30].” For easier understanding, you need to describe it with the figure.

It is hard to understand your figure and legend. Please modify them.

Author Response

Reviewer #2

General Comment: Reviewing the manuscript entitled, “Regulators of Epithelial Sodium Channels in Aldosterone-Sensitive Distal Nephrons (ASDN): Critical roles of Nedd4L/Nedd4-2 and Salt Sensitive Hypertension.” by Ishigami T. et al., this is a well-written article about relevance between Nedd4, which is an ubiquitinating enzyme, between salt sensitive hypertension. Although this is an important article for further hypertension treatment in the future, the manuscript lacks basic information of what salt sensitive hypertension is. In addition, the lack of information on renal tubular RAS makes it difficult to read at the entrance of this article.

 Comment 1: First of all, the authors should set up a paragraph on salt-sensitive hypertension and explain it to the readers of this journal.

Response 1: Thank you for your suggestion. According to the reviewer’s comments, we revised the description of salt-sensitive hypertension in the “Introduction” section, as follows:

Before: “1. Introduction

Based on data from our own[1-9] and other[10-16] studies, we previously reported that important factors in the pathogenesis of salt-sensitive hypertension include the mechanism of sodium reabsorption in the renal tubule and abnormalities that enhance such reabsorption; we also reported that angiotensinogen gene single-nucleotide polymorphisms (SNPs) can explain the pathogenesis of this disease at the levels of molecular genetics, human evolution, developmental engineering, and kidney physiology[2-4, 6, 7, 17]. In addition to the established (1) systemic renin-angiotensin system (RAS) and (2) local RAS, (3) renal tubular RAS as a third RAS might be significant in the onset of salt-sensitive hypertension. Thus, understanding renal tubular RAS and its functions is indispensable for treating salt-sensitive hypertension in humans[18]. Ultimately, diagnosing individual states of tubular RAS in each patient and identifying therapeutic procedures that could inhibit renal tubular RAS are one of our future goals of translational research in salt-sensitive hypertension. Currently, this report describes already enforced and ongoing investigations into epithelial sodium channels (ENaCs) in terminal nephrons and its regulatory ubiqutinating enzyme, Nedd4L/Nedd4-2, based on our findings.”

After: “1. Introduction

Hypertension is a multifactorial disease determined by both genetic and environmental factors including dietary habit and is a representative public health issue because of its high prevalence and the risk for cardiovascular diseases. The relationship between oral salt intake and elevation of blood pressure have been widely observed in clinical trials such as INTERSALT Study[1], PURE Study[2], and others[3]. The salt sensitivity of blood pressure is defined as “a physiological trait present in rodents and other mammals, including humans, by which the blood pressure (BP) of some members of the population exhibits changes parallel to changes in salt intake”[4]. Genetic analyses for hereditary and familial hypertension by positional cloning in the 1990’s revealed ion transporters and their accessory proteins, located in the renal tubules are responsible for development of genetic hypertension[5]. Based on data from our own[6-14] and other[15-21] studies, we previously reported that important factors in the pathogenesis of salt-sensitive hypertension include the mechanism of sodium reabsorption in the renal tubule and the abnormalities that enhance this reabsorption. Furthermore, we also reported that angiotensinogen gene single-nucleotide polymorphisms (SNPs) may explain the pathogenesis of this disease at the levels of molecular genetics, human evolution, developmental engineering, and kidney physiology[7-9, 11, 12, 22]. Currently, this report describes already enforced and ongoing investigations into epithelial sodium channels (ENaCs) in terminal nephrons and its regulatory ubiquitinating enzyme, Nedd4L/Nedd4-2, based on our findings, which might be significant molecular insight into onset and development of salt-sensitive hypertension.”

 Comment 2: In 1. Introduction, the authors emphasis the relationship between sodium reabsorption abnormality and renin angiotensin system particular renal tubular RAS for onset of salt-sensitive hypertension. However, the authors do not explain the detailed mechanism of salt-sensitive hypertension. By the way, what is renal tubular RAS? Has it reached to the consensus of all scientists? Authors should explain the phenomenon of generally lacking consensus with a figure.

Response 2: Thank you for your suggestion. We completely agreed with the reviewer regarding the excessive description of tubular renin-angiotensin system in the “Introduction” section. We have revised our manuscript as follows:

Before: “1. Introduction

Based on data from our own[1-9] and other[10-16] studies, we previously reported that important factors in the pathogenesis of salt-sensitive hypertension include the mechanism of sodium reabsorption in the renal tubule and abnormalities that enhance such reabsorption; we also reported that angiotensinogen gene single-nucleotide polymorphisms (SNPs) can explain the pathogenesis of this disease at the levels of molecular genetics, human evolution, developmental engineering, and kidney physiology[2-4, 6, 7, 17]. In addition to the established (1) systemic renin-angiotensin system (RAS) and (2) local RAS, (3) renal tubular RAS as a third RAS might be significant in the onset of salt-sensitive hypertension. Thus, understanding renal tubular RAS and its functions is indispensable for treating salt-sensitive hypertension in humans[18]. Ultimately, diagnosing individual states of tubular RAS in each patient and identifying therapeutic procedures that could inhibit renal tubular RAS are one of our future goals of translational research in salt-sensitive hypertension. Currently, this report describes already enforced and ongoing investigations into epithelial sodium channels (ENaCs) in terminal nephrons and its regulatory ubiqutinating enzyme, Nedd4L/Nedd4-2, based on our findings.”

After: “1. Introduction

Hypertension is a multifactorial disease determined by both genetic and environmental factors including dietary habit and is a representative public health issue because of its high prevalence and the risk for cardiovascular diseases. The relationship between oral salt intake and elevation of blood pressure have been widely observed in clinical trials such as INTERSALT Study[1], PURE Study[2], and others[3]. The salt sensitivity of blood pressure is defined as “a physiological trait present in rodents and other mammals, including humans, by which the blood pressure (BP) of some members of the population exhibits changes parallel to changes in salt intake”[4]. Genetic analyses for hereditary and familial hypertension by positional cloning in the 1990’s revealed ion transporters and their accessory proteins, located in the renal tubules are responsible for development of genetic hypertension[5]. Based on data from our own[6-14] and other[15-21] studies, we previously reported that important factors in the pathogenesis of salt-sensitive hypertension include the mechanism of sodium reabsorption in the renal tubule and the abnormalities that enhance this reabsorption. Furthermore, we also reported that angiotensinogen gene single-nucleotide polymorphisms (SNPs) may explain the pathogenesis of this disease at the levels of molecular genetics, human evolution, developmental engineering, and kidney physiology[7-9, 11, 12, 22]. Currently, this report describes already enforced and ongoing investigations into epithelial sodium channels (ENaCs) in terminal nephrons and its regulatory ubiquitinating enzyme, Nedd4L/Nedd4-2, based on our findings, which might be significant molecular insight into onset and development of salt-sensitive hypertension.

 Comment 3: From line 87 to line 89, the authors mentioned, “The onset of salt-sensitive hypertension caused by an abnormal renal tubular renin-angiotensin system (RAS) might be caused by abnormalities in the sodium reabsorption mechanism in the terminal nephron, that is, ASDN.” What is an abnormal renal tubular renin-angiotensin system (RAS)? First the authors need to mention about the function of normal renal tubular RAS. In addition, in the case of common cardiovascular diseases, RAS is overactive in compensation, and in chronic states, overactivity persists and exacerbates the disease development. This phenomenon should be distinguished from anomalies.

Response 3: Thank you for your comments. We agree with the reviewer’s comment and have added a description about salt-sensitive hypertension. We also completely agreed with the reviewer about excess description of tubular renin-angiotensin system in introduction section. As the reviewer commented, this may be hypothetical. Therefore, we revised the tone of the corresponding statements to present the hypothetical mechanisms as potential/suggested rather than well-established.

Before: “The onset of salt-sensitive hypertension caused by an abnormal renal tubular renin-angiotensin system (RAS) might be caused by abnormalities in the sodium reabsorption mechanism in the terminal nephron, that is, ASDN.”

After: “The onset of salt-sensitive hypertension caused by an enhanced renal tubular renin-angiotensin system (RAS) may be caused by abnormalities in the sodium reabsorption mechanism in the terminal nephron, namely ASDN.”

 Comment 4: The authors need to describe that detailed mechanisms of regulation of ENaC by the angiotensin II receptor. The authors only described it in line 141 to line 142. This is too short. In your tone of this reviewed article, RAS inhibitors appear to work on salt-sensitive hypertension. However, salt-sensitive hypertension is generally resistant to RAS inhibitors. Based on the clinical findings, you should explain that the therapeutic target for salt-sensitive hypertension is the regulation of ENaC, not the RAS inhibitor using the figure.

Response 4: Thank you for your comments and for your suggestion. The statements have been toned down in accordance, as follows:

Before: “However, from the apical side, angiotensin II generated from angiotensinogen secreted from the proximal renal tubule is thought to act on ENaC regulation via the angiotensin receptor[22-24].”

After: We have deleted this sentence since because the renin-angiotensin system in tubules is not a point of interest in our review.

 Comment 5: Although there is no particular problem was pointed out in the paragraph of Nedd4L / Nedd4-2, if this gene is a candidate for the next salt-sensitive hypertensive drug, it is preferable to describe the specific method of that strategy.

Response 5: Thank you for your comments. This will be discussed widely in a special edition entitled "Ubiquitination in Health and Diseases" which the first author of the manuscript will participate as guest editor. We raised some candidate approaches for regulating Nedd4L/Nedd4-2 about drug discovery in the “Summary” section, as follows:

“Furthermore, therapeutic and diagnostic applications targeting the NEDD4L gene should become available in the foreseeable future, including de-ubiquitination enzyme activating agents for antagonizing human NEDD4L.”

 Comment 6: Minor concerns

From line 118 to line 119, the authors mentioned “The outline of sodium transport in ASDN renal tubular epithelial cells is as shown in the reference[30].” For easier understanding, you need to describe it with the figure.

Response 6: Thank you for your comment and for bringing this issue to our attention. According to the reviewer’s comment, we have revised Figure 1.

 Comment 7: It is hard to understand your figure and legend. Please modify them.

Response 7: Thank you for your comments. We have revised the figure legends and additional figures, as requested by the reviewers.

Reviewer 3 Report

Ishigami et. al. offer review article of ENaC regulators in ASDN with focus on the roles of Need4-2 in salt-sensitive hypertension. The article in a timely contribution to the field and caused a positive impression in the reviewer. Below are some suggestions to improve its value before publication.

General comments:

The information on the article in really condensed and at sometimes hard to follow. Some extra words can be used throughout together with English language stylish.

In the introduction the authors put a special emphasis in salt-sensitive hypertension as a clinical entity and speculate to use the individual states of tubular RAS in personalized medicine. It will be interesting in this context to discuss the actual clinical efforts to segregate and give personalized treatment to patients with sensitivity to salt.

 Line 121: I recommend the use of “transmembrane Na gradient” as opposed to “intracellular”.

Need4 is introduced twice, in Section 3 and Section 4. Also it is unclear why sometimes it’s capitalized and sometimes it is not. If this is to differentiate genes and proteins I recommend clarifying when you talk about genes as mice genes names for instance are capitalized on the first letter only.  Alternatively the used of capitalization to differentiate genes and proteins can be explained in Section 11: abbreviations (and acronyms).

Lines 239-248: The link for the association of Nedd4L and essential hypertension to be a causative of salt sensitivity is unclear.

Caption from figure 1 is illegible.

Two extra figures showing the gene and protein structures of Nedd4-2 will be of great help to follow the text.

Author Response

Reviewer #3

General Comment : Ishigami et. al. offer review article of ENaC regulators in ASDN with focus on the roles of Need4-2 in salt-sensitive hypertension. The article in a timely contribution to the field and caused a positive impression in the reviewer. Below are some suggestions to improve its value before publication.

Response: Thank you for your comments, they are greatly appreciated. We revised the entire manuscript and provided an updated version with this submission.

 Comment 1: The information on the article in really condensed and at sometimes hard to follow. Some extra words can be used throughout together with English language stylish.

Response 1: Thank you for your comments. We have revised figure legends and additional figures, as requested by the reviewers, to improve the readers’ understanding. We have asked a native English editor to edit the language and grammar of our manuscript.

 Comment 2: In the introduction the authors put a special emphasis in salt-sensitive hypertension as a clinical entity and speculate to use the individual states of tubular RAS in personalized medicine. It will be interesting in this context to discuss the actual clinical efforts to segregate and give personalized treatment to patients with sensitivity to salt.

Response 2: Thank you for your recommendation. According to the reviewer’s comments, we have revised the “Introduction” section.

Before: “1. Introduction

Based on data from our own[1-9] and other[10-16] studies, we previously reported that important factors in the pathogenesis of salt-sensitive hypertension include the mechanism of sodium reabsorption in the renal tubule and abnormalities that enhance such reabsorption; we also reported that angiotensinogen gene single-nucleotide polymorphisms (SNPs) can explain the pathogenesis of this disease at the levels of molecular genetics, human evolution, developmental engineering, and kidney physiology[2-4, 6, 7, 17]. In addition to the established (1) systemic renin-angiotensin system (RAS) and (2) local RAS, (3) renal tubular RAS as a third RAS might be significant in the onset of salt-sensitive hypertension. Thus, understanding renal tubular RAS and its functions is indispensable for treating salt-sensitive hypertension in humans[18]. Ultimately, diagnosing individual states of tubular RAS in each patient and identifying therapeutic procedures that could inhibit renal tubular RAS are one of our future goals of translational research in salt-sensitive hypertension. Currently, this report describes already enforced and ongoing investigations into epithelial sodium channels (ENaCs) in terminal nephrons and its regulatory ubiqutinating enzyme, Nedd4L/Nedd4-2, based on our findings.”

After: “1. Introduction

Hypertension is a multifactorial disease determined by both genetic and environmental factors including dietary habit and is a representative public health issue because of its high prevalence and the risk for cardiovascular diseases. The relationship between oral salt intake and elevation of blood pressure have been widely observed in clinical trials such as INTERSALT Study[1], PURE Study[2], and others[3]. The salt sensitivity of blood pressure is defined as “a physiological trait present in rodents and other mammals, including humans, by which the blood pressure (BP) of some members of the population exhibits changes parallel to changes in salt intake”[4]. Genetic analyses for hereditary and familial hypertension by positional cloning in the 1990’s revealed ion transporters and their accessory proteins, located in the renal tubules are responsible for development of genetic hypertension[5]. Based on data from our own[6-14] and other[15-21] studies, we previously reported that important factors in the pathogenesis of salt-sensitive hypertension include the mechanism of sodium reabsorption in the renal tubule and the abnormalities that enhance this reabsorption. Furthermore, we also reported that angiotensinogen gene single-nucleotide polymorphisms (SNPs) may explain the pathogenesis of this disease at the levels of molecular genetics, human evolution, developmental engineering, and kidney physiology[7-9, 11, 12, 22]. Currently, this report describes already enforced and ongoing investigations into epithelial sodium channels (ENaCs) in terminal nephrons and its regulatory ubiquitinating enzyme, Nedd4L/Nedd4-2, based on our findings, which might be significant molecular insight into onset and development of salt-sensitive hypertension.

Comment 3:  Line 121: I recommend the use of “transmembrane Na gradient” as opposed to “intracellular”.

Response 3: Thank you for your comment. We have revised our manuscript as pointed out by the reviewer.

Before: “As a result of this sodium pump and K-ATPase, an intracellular sodium gradient arises, and sodium is reabsorbed through ENaCs on the apical side.”

After: “As a result of this sodium pump and K-ATPase, a transmembrane Na gradient arises, and sodium is reabsorbed through ENaCs on the apical side.”

 Comment 4: Need4 is introduced twice, in Section 3 and Section 4. Also it is unclear why sometimes it’s capitalized and sometimes it is not. If this is to differentiate genes and proteins I recommend clarifying when you talk about genes as mice genes names for instance are capitalized on the first letter only.  Alternatively the used of capitalization to differentiate genes and proteins can be explained in Section 11: abbreviations (and acronyms).

Response 4: Thank you for highlighting this issue. We have revised and differentiated the genes and proteins for both humans and mice.

 Comment 5: Lines 239-248: The link for the association of Nedd4L and essential hypertension to be a causative of salt sensitivity is unclear.

Response 5: Thank you for your comments. We have added several references for these statements.

 Comment 6: Caption from figure 1 is illegible.Two extra figures showing the gene and protein structures of Nedd4-2 will be of great help to follow the text.

Response 6: We have revised the caption of Figure 1, and have created a de novo figure, as suggested by the reviewer.

Round 2

Reviewer 1 Report

Comments have been addressed.

Reviewer 2 Report

My final decision is that this revised manuscript reaches to acceptable quality.

Reviewer 3 Report

This is a great manuscript. I have no further comments